# Emission Spectroscopy-Based Sensor System to Correlate the In-Cylinder Combustion Temperature of a Diesel Engine to NO_x_ Emissions

**DOI:** 10.3390/s24082459

**Published:** 2024-04-11

**Authors:** Jürgen Wultschner, Ingo Schmitz, Stephan Révidat, Johannes Ullrich, Thomas Seeger

**Affiliations:** 1Institute of Engineering Thermodynamics, University of Siegen, D-57076 Siegen, Germany; 2Center for Sensor Systems (ZESS), University of Siegen, D-57076 Siegen, Germany; 3Hyundai Motor Europe Technical Center GmbH, D-65428 Rüsselsheim, Germany

**Keywords:** emission spectroscopy, diesel engine, combustion temperature, NO_x_ concentration

## Abstract

Due to a rising importance of the reduction of pollutant, produced by conventional energy technologies, the knowledge of pollutant forming processes during a combustion is of great interest. In this study the in-cylinder temperature, of a near series diesel engine, is examined with a minimal invasive emission spectroscopy sensor. The soot, nearly a black body radiator, emits light, which is spectrally detected and evaluated with a modified function of Planck’s law. The results show a good correlation between the determined temperatures and the NO_x_ concentration, measured in the exhaust gas of the engine, during a variety of engine operating points. A standard deviation between 25 K and 49 K was obtained for the in-cylinder temperature measurements.

## 1. Introduction

Despite the technical development and expanding availability of renewable energies, fossil fuels still cover the majority of the energy demand. Due to the rapidly increasing global demand, especially in the mobile sector, oil requirements will slightly increase or at least remain the same by 2030 [1]. To reduce the negative effects of this development as far as possible, it is necessary to increase the efficiency of existing technologies and reduce their pollutant emissions. Typical pollutants that are formed during a combustion process are, for example, CO_2_, CO, NO_x_, and particulate matter like soot [2,3]. NO_x_ from combustion processes originates from three mean processes: thermal NO, prompt NO and fuel NO. For diesel engines, thermal NO is of special interest since it is formed from the oxidation of atmospheric nitrogen at relative high temperatures. These high temperatures are required because a very high activation energy due to the strong bond of the N_2_ molecule is necessary. The principle reactions governing the formation of NO from N_2_ were originally proposed by Zeldovich [4,5]. More details can be found in combustion-related textbooks (e.g., [6]). Using methanol or ethanol blended diesel fuels is in principle a possibility to reduce the in-cylinder combustion temperature, and as a result, the NO_x_ emissions as well [7,8,9]. In this case, the high cooling due to the enthalpy necessary for the evaporation of methanol led to lower combustion temperatures. Nevertheless, there are some disadvantages, such as the poor ignition behavior due to the low cetan number, more corrosion in the fuel supply system, and the need for additives or surfactants to produce stable alcohol blends [9].

Therefore the combustion temperature is a clear indicator for in-cylinder NO_x_ formation, and is of the utmost importance to an understanding of the NO_x_ formation process. Beside this, NO_x_ emission control also constitutes a critical issue in modern diesel engines. The in-cylinder NOx formation is typically controlled by means of Exhaust Gas Recirculation (EGR), of the internal or external type, and/or by means of a proper calibration of the injection parameters [10]. However, these techniques are typically calibrated at steady-state operation, and may be less effective in transient conditions, or at least require a high calibration effort in order to identify proper correction maps [11].

For this reason, the interest in the development of NO_x_ control mechanisms, with specific focus on transient operation, has increased [11,12]. Nevertheless, NO_x_ sensor measurements at the engine exhaust suffer from a slow response time, which is a drawback under transient conditions [13]. Model-based approaches, instead, have the potential to overcome this issue [12,14,15]. Input parameters for such approaches include, e.g., the in-cylinder pressure curve or the flame temperature [15,16].

Laser-based optical measurement techniques, e.g., Raman spectroscopy, laser induced fluorescence or laser induced gratings, offer the advantage of a nonintrusive temperature measurement, but have a significant disadvantage. When using laser systems, a complex setup has to be installed, and there is a need to modify the engine, e.g., glass windows in the piston or in the cylinder walls are necessary [17,18,19,20,21]. Previous studies show that the spark plug bore of SI-engines or the glow plug bore of diesel engines can be used to allow in-cylinder optical access without engine modifications [22,23,24,25]. Using this approach to collect spectrally resolved in-cylinder soot emissions allows global diesel engine temperature measurements. Up until now, in-cylinder pyrometric flame temperature measurements have been based on measuring the continuous black body radiation at two or several distinct wavelengths emitted by the soot [26,27,28]. This technique was also applied to two dimensional temperature measurements in an optical fully accessible engine [29,30]. Recently, this technique was extended to surface temperature measurements in non-sooting-combustion-relevant environments [31]. Nevertheless, these approaches, based on distinct wavelengths, are sensitive to wavelength selection, wavelength depending constants, soot property uncertainties and possible molecular chemiluminescence [32]. Therefore this technique was extended to spectrally resolved approaches, applied to sooting diffusion flames, and was recently used to measure temperatures during the combustion of pyrite [33,34,35]. The potential for application in a production diesel engine was shown by Block et al. by using a streak camera for time-resolved signal detection [36]. Nevertheless, in this case, the temperature was determined by using multiple distinct wavelengths.

Therefore, here, the unmodified glow plug bore of a serial engine was used to adapt an optical sensor to the combustion chamber, without any further engine modifications to detect the soot emission over a large spectral range. A serial diesel engine was examined to demonstrate that it is possible to map the correlation of the in-cylinder combustion temperature and the thermally formed NO_x_ concentration at several different engine operating points.

## 2. Setup

### 2.1. Optical Setup

The developed optical sensor is a simple modular system collecting spectrally resolved in-cylinder soot emission, which allows global diesel engine temperature measurements. The emission spectroscopy-based sensor (ESS) was designed for minimally invasive integration into the engine. For this reason, the glow plug bore of the engine is used to adapt the sensor directly to the combustion chamber. Thereby it is possible to analyze near-series diesel engines without further modifications. Figure 1a displays the optical geometry of the probe head (lower part of Figure 1a), which is almost identical to a standard glow-plug (upper part of Figure 1a), which ensures that the compression ratio is not affected. The probe head part protruding into the cylinder has a length of 79.5 mm and a tip diameter of 8.5 mm.

The light emitted during the combustion is detected in a cone of 25°, and passes the probe head through a 2.8 mm, temperature-resistant sapphire glass (Figure 1b). The heat loss properties of the combustion chamber are negligibly affected through this small window. Afterwards, the light is guided by an optical bundle, consisting of 520 single quartz fibers, optimized for the detection of UV to VIS signals. In a spectrograph equipped with a grating of 200 g/mm and a blaze at 730 nm, a spectral range between 200 nm and 850 nm can be detected. A Charge-Coupled Device (CCD) camera with a resolution of 1024 × 255 pixel converts the emitted light in an evaluable electrical signal. With this system, it is possible to detect the spectrally resolved signal in time resolutions down to 10 µs, which corresponds to 5 °CA at an engine speed of 2000 rpm. To consider the specific transmissivity of the ESS system, the measured signals are corrected with a spectral calibration curve, received by an integrated sphere, which emits light in a known spectrally resolved distribution of intensity. Figure 2 shows the complete optical setup.

### 2.2. Engine Setup

The measurements were performed in a Hyundai U2 diesel engine mounted on a test bench. The 4-cylinder in-line engine with a displacement volume of 1.6 l and European emission standard EU 5 was equipped with a common rail injection system. A constant operation point with an engine speed of 2000 rpm and an indicated mean effective pressure (IPEM) of 12 bar, along with a 125 Nm upper partial-load range, was selected for the measurements. Beside this, the other parameters are described in Section 4. The position of the crank shaft was measured with a common optical detection system, attached to the crankshaft. Aside from the temperature measurements performed using the ESS-system attached to cylinder 1, the pressure in cylinders 2, 3 and 4 was measured with piezoelectric sensors. Additionally, for all varied parameters, the NO_x_ concentration in the exhaust gas was measured with a Bosch EGS NX system. Figure 3 shows the engine test bench with the probe head and fiber connected to the engine.

## 3. Data Evaluation

The electromagnetic radiation emitted by the soot formed during a combustion process can approximately be described by Planck’s Law. This can be used in pyrometric flame temperature determination by measuring the continuous black body radiation.

In this work, the complete shape of the intensity curve in a spectral range from 300 nm to 800 nm is used to determine the combustion temperature. Since soot is not an ideal black body radiator, the specific emissivity, depending on the wavelength, must be considered [37], resulting in the spectral density distribution
(1)$$M\left(T,\lambda ,m\right)=\frac{N\cdot 8\cdot \pi^{3}\cdot D^{3}\cdot h\;\cdot \;c^{2}}{\lambda^{6}\cdot \;\left(e^{\left(\frac{h\;\cdot \;c}{\lambda \;\cdot \;k\;\cdot \;T}\right)}-1\right)}\cdot E\left(m\right)$$
and the emissivity
(2)Em=−Imm2−1m2+2
of the soot, which is a function of the complex refractive index m [38]. The parameters of Equation (1) are listed in Table 1.

The complex refractive index, a material-specific parameter of soot, was closely examined [39], and can be determined via
(3)$$m=2.213+9.551\cdot 10^{3}\;\lambda -\left(0.7528+1.265\cdot 10^{4}\lambda \right)i.$$

These fundamentals are used to calculate theoretical spectral density distributions for a wide range of temperatures. Via a contour fit procedure, using the Levenberg–Marquardt method [40], the normalized and spectrally resolved experimental signals are compared with the library of theoretical spectra, resulting in a temperature based on the best fit. Due to the normalization of the emission spectra, it is possible to neglect the size of the soot particles and the in-cylinder soot concentration. This method is well known and used in many applications (see, e.g., [41,42,43]). As an example, in Figure 4, two flame spectra are shown together with the best fitting library spectrum, resulting in temperatures of 1777 K and 2181 K.

Regarding temperature validation, the ESS system was compared to a laser-based optical measurement technique, coherent anti-Stokes Raman spectroscopy (CARS) [44]. The measurements were performed in a laminar premixed ethane/air flame established on a McKenna burner. For 100 ESS measurements, a standard deviation of 2.97 K was determined, and the mean temperature difference for both techniques was found to be about 42 K for temperatures up to 2200 K, showing a good agreement between both techniques.

## 4. Results and Discussion

A 1.6 l 4-cylinder Hyundai diesel engine was used for the ESS sensor tests. Table 2 shows the engine parameters that were varied during the measurement campaign. The different engine operation conditions are related to different NO_x_ emissions. Please keep in mind that these different engine operation points were chosen to ensure a variety of NO_x_ emissions, and not to optimize the engine performance. As a reference, standard engine conditions were used (standard ECU). The two pilot injections, which were intended to optimize the combustion and thereby lead to a noise reduction, were separately and afterwards simultaneously disabled. Further, the amount of fuel in the pilot injections was separately increased by 50%. The location of the center of combustion (CA 50), denoting the crank angle (CA) position where 50% of the injected fuel is chemically converted, was set to 2 °CA before the top dead center (BTDC) and 5 °CA after the top dead center (ATDC). The exhaust gas recirculation (EGR) rate was varied in 25% steps from 0% to 100%, whereby 100% refers to the highest amount the EGR-system can provide. The EGR leads to a lower combustion temperature, which reduces the thermally formed NO_x_.

The ESS sensor system allows for varying the exposure time of the CCD camera. In the first step, an exposure time of 45 °CA was selected, whereby the two different starting times (25 °CA and 5 °CA BTDC) were tested to ensure that the entire combustion was covered. For all engine operating points, Figure 5 shows the correlation between the determined temperatures and the NO_x_ concentrations measured in the exhaust gas for an exposure time of 45 °CA, starting at 5 °CA BTDC. The standard deviation of the obtained temperatures varies between 27 K and 45 K, which is low compared to the high combustion temperatures, up to 2200 K. The linear regression line is shown in Figure 5 together with the temperature residuals, denoted by the deviation of the determined temperatures from this line. The temperature residual is at its maximum at 145 K.

The standard correlation coefficient is calculated to be r^2^ = 0.68, which takes a value of one for a perfect positive correlation and zero if there is no linear dependency. One possible reason for this moderate correlation could be the low temporal resolution of the ESS sensor. As a result, cold areas at the beginning and the end of the combustion process are included in the measurements.

In the second step, a time resolution of 5 °CA was chosen. This setup was used to examine the combustion process in the relevant range between 20 °CA BTDC and 60 °CA ATDC, with a step width of 5 °CA. Figure 6a shows the detection windows of the ESS-sensor in relation to an in-cylinder pressure curve. It can be seen that nearly the entire combustion process, in time steps of 5 °CA, can be examined. For all time steps, a linear regression line was obtained, and the resulting temperature residuals for all engine operation points can be seen in Figure 6b. A small temperature residual can be found for the measurements starting at 15 °CA ATDC. This observation window is marked red in Figure 6, and is used further on.

The ESS sensor measurements for all engine operating points starting at 15 °CA ATDC with an exposure time of 5 °CA are shown in Figure 7. In this case, a standard deviation between 25 K 2212and 49 K was obtained for the in-cylinder temperature. The correlation coefficient increased to r^2^ = 0.71 and the maximum temperature residual decreased to 115 K, which is a reduction of about 20%. This clearly shows the potential of the ESS sensor for use in in-cylinder temperature measurements in a diesel engine, which are correlated to the in-cylinder NO_x_ formation.

In principle, there are several sources for temperature errors. Emissions from hot surfaces could interfere with the soot emissions. For engine applications, maximum wall temperatures of approximately 600 K could be expected. The resulting emission signal is several orders of magnitude lower, and has no influence on the in-cylinder temperature measurements. Chemiluminescence signals from flame front radicals or molecules like OH, CH, C_2_, CO_2_ or H_2_O are another possible source of errors. Nevertheless, the intensity of these chemiluminescence signals is too low to be observed in the spectra. The gas phase absorption of the soot emission due to H_2_O, CO_2_ formaldehyde or fuel is also not relevant, since it takes place in the mid IR region [31]. However, one possible source of temperature errors in the sensor system, which needs to be investigated, is the impact of soot depositing on the sapphire glass window of the probe head. In order to analyze this, additional measurements where performed. For this purpose, the probe head was adapted to the engine, and operated at different engine conditions and engine operation times. After each engine run, the ESS sensor head was dismounted and used to measure the temperature in a laminar diffusion flame established on a standard Bunsen burner. The probe head position relative to the conditions of the flame and the burner were the same for all measurements. Figure 8 shows soot deposition on the sapphire glass after 10 min and 60 min for the same engine conditions. The determined temperatures were 1583 K (a) and 1638 K (b), and the determined standard deviations were 15 K (a) and 12 K (b), respectively.

In the second test, the engine load was varied (10 Nm, 60 Nm and 120 Nm) at a constant speed of 1600 rpm. For each operating point, an engine run of 10 min was performed, and then the ESS sensor was dismounted to measure the temperature in the laminar diffusion flame. For all three cases, the burner conditions and the position of the sensor relative to the flame were the same. Figure 9 displays the soot deposit on the sapphire glass of the probe head. The determined temperatures were 1669 K (a), 1668 K (b) and 1708 K (c), with corresponding standard deviations of 4 K (a), 4 K (b) and 6 K (c).

The results show that the impact of soot depositing on the sapphire glass window of the probe head may lead to temperature differences of 55 K or less. This is in accordance with the results of other groups, concerning the influence of soot deposition on optical fibers (e.g., [24]).

## 5. Summary

The in-cylinder combustion temperature of diesel engines is a main parameter for the prediction of thermal formed NO_x_. Therefore, an emission spectroscopy-based sensor system (ESS sensor) was developed and used to spectrally analyze the emitted light of the in-cylinder soot. Since soot is nearly a black body radiator, a modified version of Planck’s law and a contour fit procedure were utilized to determine flame temperatures. In parallel, the NO_x_ concentration in the exhaust gas was measured. The minimally invasive ESS sensor was applied to a four-cylinder in-line diesel engine. Different operation conditions with a constant speed and load were investigated. The results show a clear correlation between the measured temperature and the NO_x_ concentration in the exhaust gas for different operation conditions, and demonstrate the potential of such a sensor system to predict and control NO_x_ formation in future. A standard deviation between 25 K and 49 K was obtained for the in-cylinder temperature measurements with a time resolution of 5 CA.

## Figures and Tables

**Figure 1 sensors-24-02459-f001:**
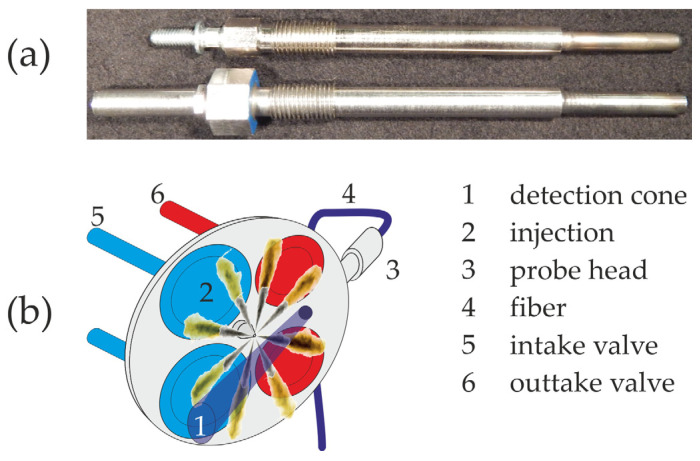
(**a**) Picture of a standard diesel engine glow-plug (**above**) in comparison to the ESS probe head (**below**). (**b**) Schematic of the cylinder head and the adapted probe head of the ESS system.

**Figure 2 sensors-24-02459-f002:**
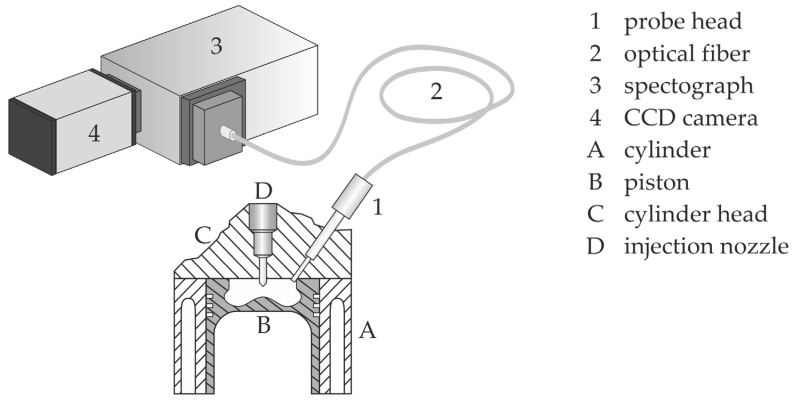
Setup scheme of the ESS system.

**Figure 3 sensors-24-02459-f003:**
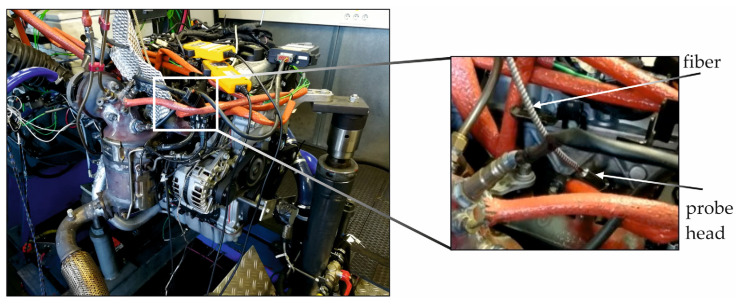
Optical probe head and fiber connected to the engine.

**Figure 4 sensors-24-02459-f004:**
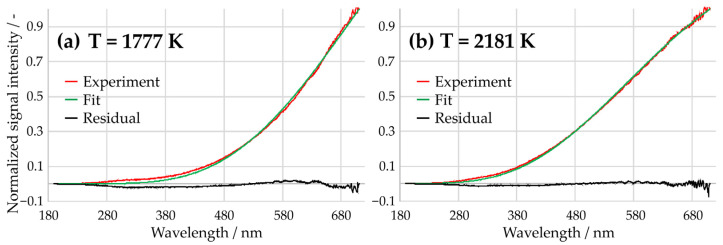
Spectrally resolved experimental signals and their best fits: (**a**) 1777 K (**b**) 2181 K.

**Figure 5 sensors-24-02459-f005:**
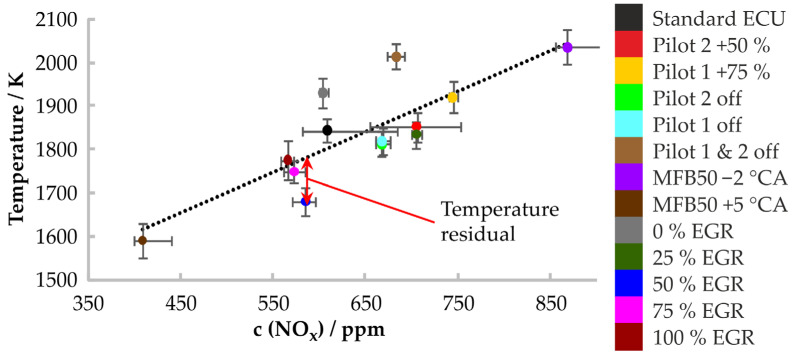
Correlation of temperature to NO_x_, measured with a resolution of 45 °CA and a starting point at 5 °CA BTDC. The grey bars represent the standard deviation of the temperatures and the NO_x_ concentrations.

**Figure 6 sensors-24-02459-f006:**
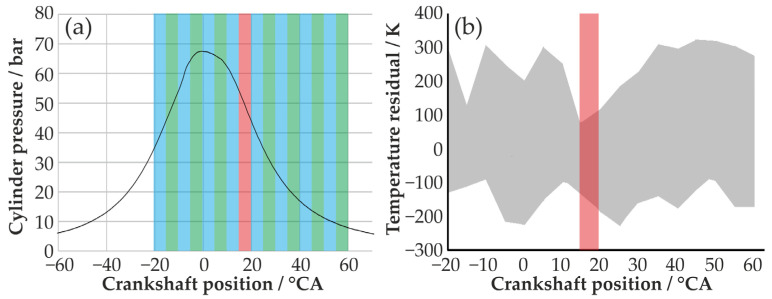
(**a**) Exposure time windows of the ESS sensor in relation to the in-cylinder pressure. (**b**) Maximum temperature residual over crankshaft position, for all engine operation points. The observation window with smallest max. temperature residual (marked red) starts at 15 °CA ATDC.

**Figure 7 sensors-24-02459-f007:**
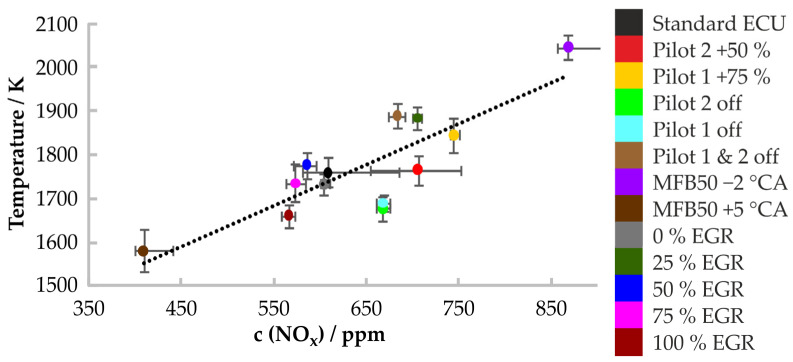
Temperature to NO_x_ correlation, measured with a resolution of 5 °CA and a starting point at 15 °CA ATDC. The grey bars represent the standard deviation of the temperatures and the NOx concentrations.

**Figure 8 sensors-24-02459-f008:**
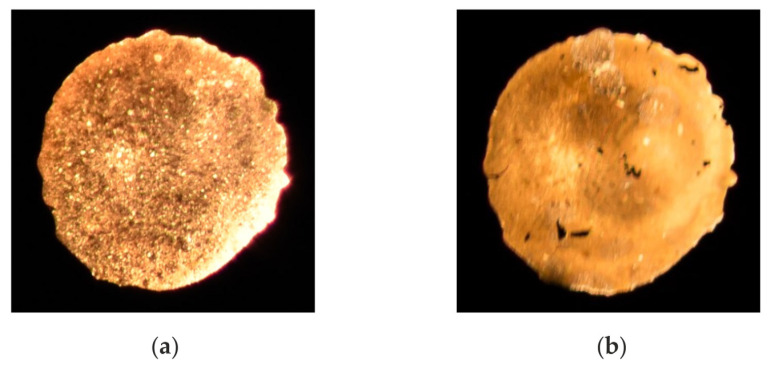
(**a**) Soot deposit after 10 min at 110 Nm and 1100 min^−1^, determined temperature 1583 K. (**b**) Soot deposit after 60 min at 110 Nm and 1100 min^−1^, determined temperature 1638 K.

**Figure 9 sensors-24-02459-f009:**
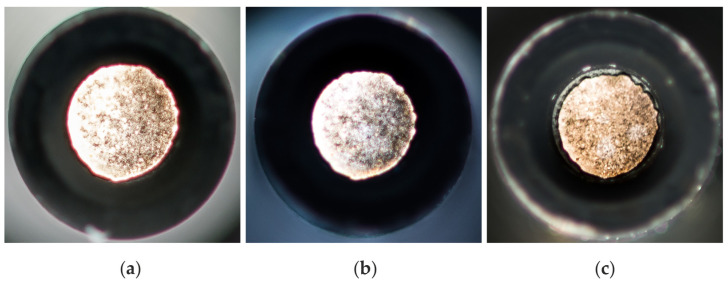
(**a**) Soot deposit after 10 min at 10 Nm and 1600 rpm, determined temperature 1669 K. (**b**) Soot deposit after 10 min at 60 Nm and 1600 rpm, determined temperature 1668 K. (**c**) Soot deposit after 10 min at 120 Nm and 1600 rpm, determined temperature 1708 K.

**Table 1 sensors-24-02459-t001:** Parameters of the spectral density distribution $M\left(T,\lambda ,m\right)$.

Parameter	Description	Unit
T	Temperature	K
λ	Wavelength	nm
N	Number of soot particles	-
D	Soot particle diameter	nm
c	Speed of light	m/s
h	Planck constant	J s
k	Boltzmann constant	J/K

**Table 2 sensors-24-02459-t002:** Engine operating points.

Engine Parameter	Settings
Pilot injection	Pilot 1 off
	Pilot 2 off
	Pilot 1 + 2 off
	Pilot 1 + 50%
	Pilot 2 + 50%
CA 50	2 °CA BTDC
	5 °CA ATDC
EGR rate	0%
	25%
	50%
	75%
	100%

## Data Availability

The raw data supporting the conclusions of this article will be made available by the authors on request.

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
