# Peer review of "Emission Spectroscopy-Based Sensor System to Correlate the In-Cylinder Combustion Temperature of a Diesel Engine to NOx Emissions"

_sensors, 2024, doi:10.3390/s24082459_

Round 1

Reviewer 1 Report

Comments and Suggestions for Authors

Title: Emission spectroscopy-based sensor system to correlate the in-cylinder combustion temperature of a diesel engine to NOx emissions

·       Abstract and conclusions.  it is recommended to include the experimental results rather than a generalized statement in the abstract as well as in the conclusions.  

·       High-temperature combustion can also be controlled with the implementation of additives like alcohols. It is suggested to authors to discuss low-temperature combustion with additives like ethanol and methanol.

·       When correlating diesel engine combustion temperature with NOx formation. It is recommended to include a better discussion of the Zeldovich mechanism. Many researchers failed/neglected to explain this mechanism.

·       It is stated that the glow plug bore of the engine is used to adapt the sensor directly to the combustion chamber. Kindly provide the technical specifications of the sensor.  

·       The CCD camera means charge-coupled device.?

·       Line 90, Page.3. The 1.6 l 4-cylinder in-line engine. The specifications are a little bit confusing. Kindly reframe the specifications.

·       Line 92, Page.3. I think it is unnecessary to underline the indicated mean effective pressure. Similar to CA, BTDC, ATDC, EGR etc.

·       Line 130.  Apart from the Levenberg-Mar-quardt method, Particle Swarm Optimization, Genetic Algorithms, and Quasi-Newton Methods are also popular for this application. Any specific reason for choosing the Levenberg-Mar-quardt method.?

·       Kindly explain the engine loading conditions when performing the experiments. 

Reviewer 2 Report

Comments and Suggestions for Authors

Pollutant emissions are a hot topic related to energy and the environment. This manuscript proposes a sensing device based on emission spectroscopy for studying the relationship between cylinder temperature and nitrogen oxide emissions in diesel engines. The manuscript is recommended to be accepted for publication after clarifying the following comments.

1.The Introduction section should further expand the current relevant progress of the research content to highlight the advantages and necessity of the proposed method in this work.

2.How to ensure the stability of spectral acquisition during the operation of the device shown in Figure 2?

3. In the Results and Discussion section, further discussion is needed regarding data analysis. Further optimize the operating conditions of diesel engines by obtaining the relationship between cylinder temperature and nitrogen oxide emissions.

4.The type of manuscript is communication, but the content seems to be too much. Please refer to the requirements for writing communication manuscripts in this journal.

5. It seems that the references need to be updated because there are too many old references cited.

Comments on the Quality of English Language

Minor editing of English language required.

Reviewer 3 Report

Comments and Suggestions for Authors

       In this paper, a sensor system based on emission spectrum is proposed to measure the combustion temperature inside a diesel engine, and its relationship with NOx emissions is discussed. This research method is novel and has certain theoretical and practical significance for reducing pollutant emissions of diesel engines. I recommend accept it after revision, the reasons are followed:

1.      Although the authors provide detailed descriptions of the experimental design and data analysis, there is a lack of comprehensive discussion on experimental conditions, sensor precision, and potential sources of error. It is recommended to include such discussions.

2.      The interpretation of certain figures and the data analysis could be further enhanced. The authors are recommended to add more descriptive labels and explanatory text to help readers better understand the data.

3.      The article's interpretation of experimental data is somewhat reasonable, but explanations for certain key findings seem oversimplified. The authors should consider including more detailed analysis on how conclusions are drawn from the data, especially when dealing with data variability and potential outliers.

4.      The Introduction would benefit from a more comprehensive literature review, some other reported emission spectroscopy sensors could be included, like DOI: 10.1364/OE.443732.

5.      Suggestion: The mechanism of the relationship between temperature and NOx emission can be further explored.
